# Ethnic Disparities in Lipid Metabolism and Clinical Outcomes between Dutch South Asians and Dutch White Caucasians with Type 2 Diabetes Mellitus

**DOI:** 10.3390/metabo14010033

**Published:** 2024-01-03

**Authors:** Lushun Yuan, Aswin Verhoeven, Niek Blomberg, Huub J. van Eyk, Maurice B. Bizino, Patrick C. N. Rensen, Ingrid M. Jazet, Hildo J. Lamb, Ton J. Rabelink, Martin Giera, Bernard M. van den Berg

**Affiliations:** 1Einthoven Laboratory for Vascular and Regenerative Medicine, Leiden University Medical Center, 2333 ZA Leiden, The Netherlands; l.yuan@lumc.nl (L.Y.); p.c.n.rensen@lumc.nl (P.C.N.R.); a.j.rabelink@lumc.nl (T.J.R.); 2Department of Internal Medicine, Division of Nephrology, Leiden University Medical Center, 2333 ZA Leiden, The Netherlands; 3Center for Proteomics and Metabolomics, Leiden University Medical Center, 2333 ZA Leiden, The Netherlands; a.verhoeven@lumc.nl (A.V.); n.blomberg@lumc.nl (N.B.); m.a.giera@lumc.nl (M.G.); 4Department of Internal Medicine, Division of Endocrinology, Leiden University Medical Center, 2333 ZA Leiden, The Netherlands; h.j.van_eyk@lumc.nl (H.J.v.E.); i.m.jazet@lumc.nl (I.M.J.); 5Department of Radiology, Leiden University Medical Center, 2333 ZA Leiden, The Netherlands; m.b.bizino@lumc.nl (M.B.B.); h.j.lamb@lumc.nl (H.J.L.)

**Keywords:** lipidomics, Dutch South Asian, Dutch white Caucasian, type 2 diabetes mellitus, diabetic nephropathy

## Abstract

Type 2 diabetes mellitus (T2DM) poses a higher risk for complications in South Asian individuals compared to other ethnic groups. To shed light on potential mediating factors, we investigated lipidomic changes in plasma of Dutch South Asians (DSA) and Dutch white Caucasians (DwC) with and without T2DM and explore their associations with clinical features. Using a targeted quantitative lipidomics platform, monitoring over 1000 lipids across 17 classes, along with ^1^H NMR based lipoprotein analysis, we studied 51 healthy participants (21 DSA, 30 DwC) and 92 T2DM patients (47 DSA, 45 DwC) from the **MAGN**etic resonance **A**ssessment of **VICTO**za efficacy in the **R**egression of cardiovascular dysfunction in type 2 d**IA**betes mellitus (MAGNA VICTORIA) study. This comprehensive mapping of the circulating lipidome allowed us to identify relevant lipid modules through unbiased weighted correlation network analysis, as well as disease and ethnicity related key mediatory lipids. Significant differences in lipidomic profiles, encompassing various lipid classes and species, were observed between T2DM patients and healthy controls in both the DSA and DwC populations. Our analyses revealed that healthy DSA, but not DwC, controls already exhibited a lipid profile prone to develop T2DM. Particularly, in DSA-T2DM patients, specific lipid changes correlated with clinical features, particularly diacylglycerols (DGs), showing significant associations with glycemic control and renal function. Our findings highlight an ethnic distinction in lipid modules influencing clinical outcomes in renal health. We discover distinctive ethnic disparities of the circulating lipidome and identify ethnicity-specific lipid markers. Jointly, our discoveries show great potential as personalized biomarkers for the assessment of glycemic control and renal function in DSA-T2DM individuals.

## 1. Introduction

One of the major challenges to public health in the twenty-first century is the worldwide rise in type 2 diabetes mellitus (T2DM) prevalence. T2DM is characterized by insulin resistance and insufficient compensatory insulin secretion, the mechanism of which varies by ethnicity [1]. South Asians (SAs), as one of the high-risk populations, have a higher T2DM incidence than other ethnic groups [2]. As a result the South Asian (SA) population with T2DM tend to develop the disease at an earlier age, around 5–10 years ahead, and often with a lower body mass index (BMI), compared to white Caucasians (wC), thus revealing a distinct disease phenotype [2]. SAs possess a distinct body composition characterized by a higher prevalence of abdominal obesity and a larger proportion of visceral fat [3]. This unique phenotype contributes to the production and secretion of specific inflammatory cytokines, which can result in an elevated chronic low-grade inflammatory state and increasing the risk of developing T2DM among this population [4,5]. Furthermore, SA patients with T2DM were found to be more prone to develop microvascular complications such as diabetic nephropathy (DN), as well as progressing to end-stage renal disease at a faster rate than Caucasian European patients with T2DM [6,7].

While emerging lipidomic approaches generally revealed specific molecular lipid changes leading to T2DM (6–8), most of these studies were only performed in single ethnicity lacking differential information on T2DM development between different ethnic groups. Of note, a number of epidemiological studies highlighted the role of dyslipidemia in relation to the incidence of T2DM [8,9], hinting that development of dyslipidemia may be a sign of future T2DM. Additionally, several studies found that dyslipidemia was associated with an increased risk of diabetes-related microvascular complications such as nephropathy, neuropathy, and retinopathy [10,11,12]. Given that dyslipidemia patterns differ by race/ethnicity [13] and may influence disease outcome [14], this suggests that lipid metabolism may play a vital role in ethnic differences in risk and progression of T2DM. In the present study, we measured lipidomic phenotypes in Dutch South Asian (DSA) and Dutch white Caucasian (DwC) participants, with or without T2DM, using the differential mobility mass spectrometry (DMS/MS)-based Shotgun Lipidomics Assistant (SLA) platform [15]. Based on this platform, we sought to investigate differences in lipid class and lipid species correlating with disease risk and progression between these two ethnic groups.

## 2. Materials and Methods

### 2.1. Study Population

For the present study, baseline samples were used from the MAGNetic resonance Assessment of VICTOza efficacy in the Regression of cardiovascular dysfunction in type 2 dIAbetes mellitus (MAGNA VICTORIA) cross-sectional study from two previous randomized controlled trials (RCT, ClinicalTrials.gov [NCT01761318] [16] and [NCT02660047] [17], respectively), together with age and gender matched healthy controls from both ethnic groups [18]. The details of both trials can be found elsewhere [16,17]. Both trials had the following inclusion criteria: BMI ≥ 23, age between 18 and 74 years, and glycated hemoglobin HbA1c levels between 6.5% and 11.0% (≥47.5 and ≤96.4 mmol/mol). Patients were allowed to take specific glucose-lowering medication (metformin, sulfonylurea derivatives, or insulin) at a stable dosage for at least 3 months prior to participating in the study. They could also use antihypertensives and statins. Exclusion criteria included the use of glucose-lowering medication other than those specified, pre-existing renal diseases of non-diabetic origin, congestive heart failure (NYHA class III-IV), uncontrolled hypertension (systolic blood pressure > 180 mmHg and/or diastolic blood pressure > 110 mmHg), or recent acute coronary or cerebrovascular events within 30 days before study enrolment. We excluded samples with missing plasma, diagnosed with T1DM, and individuals who withdraw from the randomized clinical trial. In total, 47 DSA with T2DM (DSA-T2DM, age 54.9 [SD: 10.1] years, 59.6% women, BMI: 29.5 [4.0] kg/m^2^), 21 DSA healthy individuals (DSA-C, age 48.3 [SD: 8.1] years, 71.4% women, BMI: 23.5 [3.0] kg/m^2^), 45 DwC with T2DM (DwC-T2DM, age 59.0 [SD: 6.5] years, 44.4% women, BMI: 32.3 [3.9] kg/m^2^), and 30 DwC healthy individuals (DwC-C, age 57.9 [SD: 7.9] years, 46.7% women, BMI: 24.3 [3.3] kg/m^2^) were included (Appendix A). Ethnicity was based on the self-identified and self-reported biological parents’ and ancestors’ origins. Participants with complete informed consent were included. The study was conducted in accordance with the revised Helsinki Declaration, and the Institutional Review Board granted ethical approval (Leiden University Medical Center, Leiden, The Netherlands).

### 2.2. Lipidomics Profiling Using the SLA Platform

Plasma samples were prepared according to Ghorasaini et al. [19], and analyzed on the Shotgun Lipidomics Assistant (SLA) platform (Figure 1). The SLA consists of a SCIEX QTRAP 5500 mass spectrometer with a SelexION differential mobility spectroscopy (DMS, Sciex LLC, Framingham, MA, USA) interface and a Nexera X2 ultrahigh-performance liquid chromatography system that is controlled by the SLA software (version 1.3; https://github.com/syjgino/SLA/releases). Detailed protocols on its operation can be found elsewhere [15,19]. Based on the SLA platform, we could generate 17 lipid classes including cholesteryl ester (CE), ceramide (CER), diacylglyceride (DG), dihydroceramide (DCER), fatty acid (FA), hydroxyceramide (HexCER), lactosylceramide (LacCER), lysophosphatidylcholine (LPC), lysophosphatidylethanolamine (LPE), phosphatidic acid (PA). phosphatidylcholine (PC), phosphatidylethanolamine (PE), phosphatidylinositol (PI), phosphatidylserine (PS), sphingomyelin (SM), triglyceride (TG).

### 2.3. ^1^H Nuclear Magnetic Resonance (NMR) Spectroscopy Measurement and Processing

Sample preparation was performed consistently with the requirements of the Bruker B.I.LISA lipoprotein analysis protocol as detailed in our previous study [20]. Lipoprotein values were extracted from the NOESY1D plasma spectra employing the Bruker IVDr Lipoprotein Subclass Analysis (B.I.LISA) platform [21,22,23,24,25,26]. This approach extracts information about lipoproteins and lipoprotein subfractions in plasma.

### 2.4. Statistical Analyses

For the SLA data pre-processing we first calculated the missing values per lipid class for all individuals per group (DSA-T2DM, DSA-C, DwC-T2DM, and DwC-C). Per lipid class, specific lipids with more than 30% missing values in each group were excluded. Missing values were imputed with half of the minimum concentration per lipid class (Appendix A).

Next, to determine the relative abundance of each lipid class, we performed a calculation by normalizing the concentration of each lipid class. This involved summing up the concentrations of all lipid species within each class and dividing it by the total concentration across all lipid classes. By applying this normalization process, a more accurate understanding of how each lipid class contributes to the overall lipid composition is obtained and it allows clear assessment of the proportional representation of different lipid classes. Subsequently, principal component analysis (PCA) and hierarchical cluster analysis (HCA) were performed in all participants in both ethnicities based on relative lipid class abundance. Differences in relative lipid class abundance between healthy controls and T2DM, as well as between healthy individuals and T2DM from two ethnic groups were examined.

Multinomial logistic regression analysis (MLR) was used to differentiate the various specific lipids. The four groups were considered as outcomes (DSA-T2DM, DSA-C, DwC-T2DM, and DwC-C). The lipid concentrations were scaled (z-score normalization). When comparing DSA-T2DM to DSA-C, we used DSA-C as reference, and when comparing DWC-T2DM to DwC-C, we used DwC-C as reference. Age (continuous variable), sex (dichotomous variable), and current smoking status (dichotomous variable) were adjusted for the complete model. Multiple testing corrections were used, with a false discovery rate (FDR) of 0.05 considered significant. To assess the relationship between lipid concentrations and T2DM, the results were expressed as a regression coefficient (β) with a 95% confidence interval (CI).

For weighted correlation network analysis, the “WGCNA” R package was used to investigate the role of lipid species in association with observed clinical features [27]. Using this algorithm, a proper soft threshold was first chosen and lipids with similar concentration patterns could be grouped into multiple modules tagged with multiple colour codes, each of which was linked to a concomitant clinical feature. This makes it possible to identify some clinically relevant lipids within potential lipid modules in relation to respective clinical features. Diabetes-related complications-associated lipid modules were considered key modules in this study. Key mediatory lipids that correspond to the clinical parameters were derived from key lipid modules and the differentiated lipids (between the various groups, Appendix A). Pearson’s correlation analysis was used to determine the relationship between those key mediatory lipids and clinical parameters related to dyslipidaemia, kidney function, and glycemic control.

To validate our observations, we used a published external dataset of Chinese IgA nephropathy patients to investigate the relationship between commonly changed lipids and renal function [28]. To this end, we first examined the changes in these lipids between healthy controls and IgA nephropathy patients. Next, the relationship between these lipids and kidney function parameters was determined using Pearson’s correlation analysis.

To assess the associations between key mediatory lipids and lipoprotein and lipoprotein subfractions, Pearson’s correlation analysis was performed in the diabetic populations of both ethnic groups.

For each ethnic group, the Wilcoxon signed-rank test was used to assess the statistical differences between cases (i.e., those with diabetes-related complications) and controls (i.e., those without diabetes-related complications). R (version 4.1.0) and GraphPad Prism version 8 (Graphpad Inc., La Jolla, CA, USA) were used for statistical analysis.

## 3. Results

### 3.1. Pre-Processing of Plasma Lipidome Profiles of Individuals with T2DM vs. Healthy Participants

Targeted lipidomic analysis quantified lipids from 17 different lipid classes (Figure 1). After exclusion of lipids with 30% missing values, we distinguished 689, 686, 679, 699, and 668 lipids in DSA-T2DM, DSA-C, DwC-T2DM, and DwC-C, respectively (Appendix A), of which 654 common lipid species across lipid classes (CE, cholesteryl ester; CER [Cer d18:1/FA], ceramide; DG, diacylglyceride; DCER [Cer d18:0/FA], dihydroceramide; FA, fatty acid; HexCER, hydroxyceramide; LacCER, lactosylceramide; LPC, lysophosphatidylcholine; LPE, lysophosphatidylethanolamine; PA, phosphatidic acid; PC, phosphatidylcholine; PE, phosphatidylethanolamine; PI, phosphatidylinositol; PS, phosphatidylserine; SM, sphingomyelin; TG, triglyceride) were chosen for further analysis (Appendix A).

### 3.2. Healthy Individuals of Dutch South Asian Ethnicity Reveal a pre-Diabetes Lipid Class Profile

We first applied PCA analysis to identify clusters of subjects based upon similarities in their relative abundance of lipid classes in both ethnic groups, regardless of their prespecified group. Although the distinction was not perfect, we observed that relative lipid class abundance had better power to distinguish patients with T2DM from healthy controls in DwC than in DSA (Appendix A). Hierarchical cluster analysis revealed that in DSA, all subjects were clustered into two main subclusters, one with patients with T2DM only and the other with patients with T2DM and healthy individuals; whereas in DwC, most of the healthy subjects were clustered together and separated from patients with T2DM (Figure 2). Between DSA-T2DM and DSA-C, 10 lipid classes (9 lower and 1 higher), and between DwC-T2DM and DwC-C, 11 lipid classes (10 lower and 1 higher) were changed significantly (Appendix A). When DSA-T2DM and DwC-T2DM were compared, the DG lipid class was found to be more abundant, with the greatest relative difference, in DSA with T2DM than in DwC with T2DM (Appendix A). Similarly, we also found that the abundance of DG lipid class was higher in DSA than DwC among healthy individuals (Appendix A). These findings indicated that based on lipid class abundance, DSA-C already had a phenotype more closely related to DSA-T2DM; there were marginal differences in lipid class abundance between T2DM in the two ethnic groups.

### 3.3. Comparison of Differential Lipids between Patients with T2DM and Healthy Controls in Two Ethnicities

After multinomial logistic regression analyses and multiple testing corrections, we found 436 differential lipids (396 higher and 40 lower) in DSA-T2DM compared to DSA-C (Figure 3A and Appendix A); 519 differential lipids (471 higher and 48 lower) in DwC-T2DM compared to DwC-C (Figure 3A and Appendix A). To further investigate the significance of each lipid change between two ethnicities, we compared the regression coefficients (T2DM vs healthy controls) and discovered that lipids from the DGs and TGs classes in DSA showed higher regression coefficients than those in DwC, while the CEs in DwC showed lower regression coefficients; the remaining lipids behaved similarly (Figure 3A).

We found 9 lipids that were specifically lower in DSA-T2DM (mostly from the CEs and LPCs) and 17 lipids that were specifically lower in DwC-T2DM (including lipids from the FAs, PCs, SMs, and TGs) (Figure 3B,D, and Appendix A). Furthermore, 13 lipids were specifically higher in DSA-T2DM (primarily from the DGs, PEs, and TGs), while 88 lipids were specifically higher in DwC-T2DM (primarily from the CEs, DGs, PCs, PEs, and TGs) (Figure 3C,E and Appendix A). These findings indicate that different lipid metabolism phenotypes were found in both ethnicities, and differential lipids, particularly DGs and TGs, showed the greatest associations with the risk of T2DM in DSA.

### 3.4. Ethnic Distinction in Associations of Lipid Correlation Network Modules with Clinical Features

To identify highly connected lipid modules and the relevance between baseline clinical traits and each lipid module, we performed a weighted correlation network analysis (WGCNA, Appendix A). Except for the grey module, which corresponded to the set of lipids that were not clustered in any module, the other lipids in both ethnicities were clustered into 12 modules.

Total TG concentration measured on the SLA platform matched the clinical routine measurements. In both ethnic groups, we did observe lipid modules consisting of TG species that had a positive correlation with total TG concentration (Figure 4 and Appendix A). Surprisingly, there were noticeable differences between the two ethnic groups. In DSA-T2DM, the lipid modules were positively correlated with glycemic control parameters, and negatively correlated with high-density lipoprotein (HDL)-cholesterol (Figure 4A and Appendix A). While in DwC-T2DM, the lipid modules showed positive correlations with anthropometric parameters, total cholesterol, and low-density lipoprotein (LDL)-cholesterol and negative correlations with blood pressure and kidney function (Figure 4B and Appendix A). We also found several modules associated with diabetes-related complications. In DSA-T2DM, the ‘royal blue’ module, as only module, was correlated with diabetic nephropathy (DN); whereas in DwC-T2DM, the ‘light green’, ‘black’, and ‘grey60’ modules were associated with diabetic retinopathy (DR) and DN, respectively (Figure 4 and Appendix A).

By combining lipids in diabetes-related-complications modules with differential lipids, we identified 7 lipids from the DG class (two lipids showed ethnicity-specific difference; DG 16:0_18:2 and DG18:1_18:2) which were specifically higher in DSA and 5 lipids from the CEs, TGs, and DG classes which were lower in DwC (Appendix A). These lipids were considered as key mediatory lipids. Our findings revealed ethnic differences in the associations between lipid modules and clinical features, particularly in DSA-T2DM, where lipid modules were correlated with high TGs, low HDL-cholesterol, and poor glycemic control.

### 3.5. Clinical Relevance Screening for Key Mediatory Lipids from Two Ethnicities

The key mediatory lipids of DSA were first investigated in relation to clinical features such as dyslipidaemia, kidney function, and glycemic control parameters in both ethnicities. These lipids correlated positively with total TGs, total cholesterol, albumin/creatinine ratio, and HbA1c in DSA-T2DM, but negatively with HDL-cholesterol and LDL-cholesterol (Figure 5A and Appendix A). Since these lipids were derived from a DN-related module, we next compared the lipid concentrations in patients with and without DN. All the lipids, except DG 18:2_20:4, were higher in DN than T2DM in DSA (Figure 5B). However, we found only a few correlations between these lipids and LDL-cholesterol in DwC-T2DM (Figure 5C and Appendix A). In DwC, between T2DM and DN, those lipids exhibited the opposite behaviour (Figure 5D).

Key mediatory lipids derived from modules in DwC were then examined. Only limited correlations with clinical parameters could be observed in both ethnic groups (Appendix A). None of them showed associations with DR or DN in either ethnicity (Appendix A). These findings suggested that DGs were more strongly associated with DSA-T2DM than DwC-T2DM and with DN and kidney function.

We finally investigated key mediatory lipids of DSA in an external Chinese cohort of patients with IgA nephropathy and found that they were all higher in patients with IgA nephropathy than in healthy controls, with DG 18:1_18:2 showing the strongest correlation with renal function parameters (Appendix A).

## 4. Discussion

In the current lipidomic phenotyping study, we discovered differences in lipid classes and lipid species between patients with T2DM and healthy individuals in both the Dutch South Asian (DSA) and Dutch white Caucasian (DwC) populations. Specifically, lipid changes in individuals with T2DM of DSA were found to be more strongly associated with clinical parameters than DwC, with diacylglycerols (DGs) showing strong associations to diabetic nephropathy and renal function. Furthermore, we observed that healthy DSA individuals already had a diabetes-prone lipid distribution. These findings imply that impaired DG metabolism in DSA could be a potential hallmark and that lipidomic phenotyping could provide detailed insights into lipid metabolic complexity and interindividual variations among T2DM patients of various ethnic groups.

Previous studies suggested that SAs may have a lower ability to secrete insulin, lower muscle mass and a higher ectopic fat deposition which contributed to the higher T2DM prevalence [3,29]. In the current study, it is worth noting that healthy DSA individuals already revealed a diabetic lipid distribution, which partly could predict the higher risk in developing T2DM in this population. Additionally, our study revealed remarkable differences in lipidomic profiles between both ethnic groups, lending credence to previously established associations between T2DM and dysregulated lipoprotein composition using ^1^H NMR lipoprotein profiling [20]. Comparing DG 18:1_18:2 to these plasma lipoprotein profiling data in DSA-T2DM (Appendix A) revealed positive correlations with numerous of these markers while in DwC-T2DM associations with all HDL subfractions, most IDL subfractions, and a few LDL subfractions were absent. Our results demonstrate distinct differences in the lipidome between patients with T2DM and healthy controls, mainly related to CE, DG, PE, SM, and TG metabolism. This was consistent with previous findings observed in the case-cohort study nested within the PREDIMED (PREvención con DIeta MEDiterránea) trial [30] and the longitudinal METSIM (METabolic Syndrome In Men) study [31]. However, conflicting results were reported in two studies based on Chinese populations [32,33]; for instance, compared to healthy controls, free-fatty acids (FFA), sphingomyelin (SM) and lysophosphatidylcholine (LPC) lipid species were higher in Chinese patients with T2DM, whereas we found opposite results in Dutch patients with T2DM, further highlighting the variability in lipidomics profile between ethnicities.

The comprehensive analysis for lipidomics profiling performed in the current study allows for testing clinically relevance. As a hallmark of T2DM, insulin resistance affects regulation of lipid and lipoprotein metabolism [34,35]. In line with previous studies, we found that lipid modules in DSA-T2DM positively correlated with TG, total cholesterol and negatively correlated with HDL-C; whereas lipid modules in DwC-T2DM correlated with total TG, cholesterol, and LDL-C, suggesting an ethnic preference in correlation with dyslipidemia patterns. Insulin resistance also impairs glucose metabolism and TG metabolism [36,37,38]. Interestingly, we discovered ethnicity differences in lipid modules (mainly consisting of TGs and DGs) demonstrating a correlation with both short- and long-term glycemic control exclusively in DSA-T2DM, rather than in the DwC-T2DM population. Also, our observation that certain lipid modules correlated with DN in both ethnic groups was in line with the reported dyslipidemia as a hallmark of chronic kidney disease (CKD) [39]. Moreover, a previous study revealed that patients with CKD had abnormalities in glycerolipid metabolism such as monoradylglycerolipids (MG), DGs, and TGs [40], which is also consistent with our findings in DSA-T2DM. However, we did not observe these associations in DwC-T2DM; one possible hypothesis might be a shorter duration in diabetes, which resulted in the more excessive changes in lipid metabolism.

By combining lipid abundance and lipid species analysis, we have identified a specific lipid class, DG, which is associated with the increased risk in development and progression of T2DM among SAs. DG is derived from lipoprotein lipase (LPL)-mediated hydrolysis of TGs, and our observations reveal low DG abundance alongside high TG levels in T2DM in both ethnicities. However, DG lipid class abundance was much higher in DSA than DwC in both healthy and diabetic individuals, hinting to possible lipolysis dysregulation in SAs. Insulin resistance, a crucial factor in T2DM development, has been found to be higher in SAs than in wCs [41,42]. LPL has been associated with insulin resistance [43,44], and this could potentially explain the higher proportion of DG observed in our study, as higher insulin resistance in SAs impairs the ability of insulin to suppress lipolysis, leading to an increased release of fatty acids that are subsequently converted to DGs. Previous research has reported the impact of DGs on hepatic insulin resistance. Increased levels of DGs are commonly observed in animal models of lipid-induced hepatic insulin resistance [45,46]. Furthermore, several human studies have demonstrated significant associations between total hepatic DG content or specific DG species and insulin resistance markers, such as homeostasis model assessment-estimated insulin resistance (HOMA-IR) [47,48,49,50]. These associations were found to be stronger than those observed with variables like body mass index, ceramide content, and markers of endoplasmic reticulum stress [47]. Interestingly, DGs and their targets protein kinase C (PKC) and protein kinase D (PKD) have been shown to regulate multiple critical cellular responses [51], which might be a plausible mechanism for inhibition of insulin signalling leading to hepatic insulin resistance. Once DG accumulates, it could lead to hyperactivation of PKC/PKD and play an important role in development of diabetic nephropathy [52,53]. Our observation that DG metabolism in the circulation was disturbed, with a higher correlation to clinical outcomes, may argue that a dysregulated DG-PKC/PKD signalling network could disrupt the redox balance and lead to more oxidative stress [53], and in part could explain why DSA-T2DM patients are more vulnerable to diabetic nephropathy progression.

The strength of our study is that we measured detailed lipidomic profiles in two ethnic groups of diabetic and healthy individuals. Our findings confirmed lipidomic perturbations in patients with T2DM in both ethnic groups; meanwhile, we revealed an ethnic distinction of lipid modules in relation to clinical outcomes (e.g., glycemic control). Notably, there are still several limitations to our study. First, our study is a cross-sectional study; therefore, we cannot address issues of causality in the association of T2DM. Second, the relatively small sample size limits the power of generalization and precludes stratification analyses. Third, as waist-to-hip ratio (WHR) was found to be the most reliable predictor of T2DM in the HELIUS (HEalthy LIfe in an Urban Setting)study, regardless of ethnicity [54], lack of a WHR matching design might be a shortcoming in our study. Fourth, oxidized lipids induced by oxidative stress play a critical role in the development and progression of T2DM [55,56], however, we were not able to detect oxidized lipids using this high-throughput platform. Fifth, for renal function validation we only used an external cohort of patients with IgA nephropathy instead of diabetic nephropathy, and in a singular ethnic group. Therefore, further longitudinal studies with multiple ethnic groups and larger sample sizes are needed to verify our findings. Sixth, it is important to consider the influence of dietary factors on lipid profiles [57,58,59], especially since substantial differences in dietary patterns have been observed between Dutch South Asians and Dutch white Caucasians [60]. However, our study did not gather specific information regarding the participants’ dietary patterns.

## 5. Conclusions

In conclusion, Dutch patients with T2DM of both white Caucasian and South Asian descent exhibited altered circulating lipidomes when compared to healthy individuals of the same ethnicity. In DSA the lipid changes of especially DGs, were clinically more relevant than in DwC. These DGs, particularly DG 18:1_18:2, were associated with glycemic control and renal function in DSA patients with T2DM and Chinese patients with IgA nephropathy (validation cohort). These observations suggest that they could be used as ethnicity-specific biomarkers for diabetic nephropathy patients. In addition, lipidomics phenotyping provides detailed insight into lipid metabolic complexity and interindividual variations among patients with T2DM from various ethnic groups.

## Figures and Tables

**Figure 1 metabolites-14-00033-f001:**
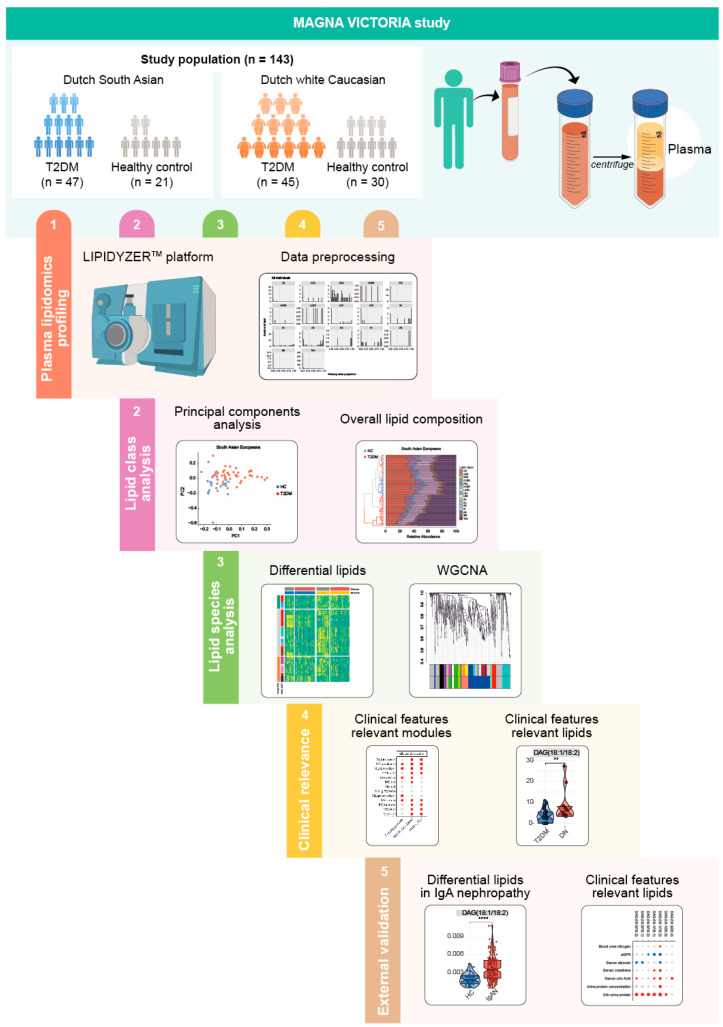
Study workflow and design. Abbreviations: T2DM type 2 diabetes mellitus; WGCNA Weighted Correlation Network Analysis.

**Figure 2 metabolites-14-00033-f002:**
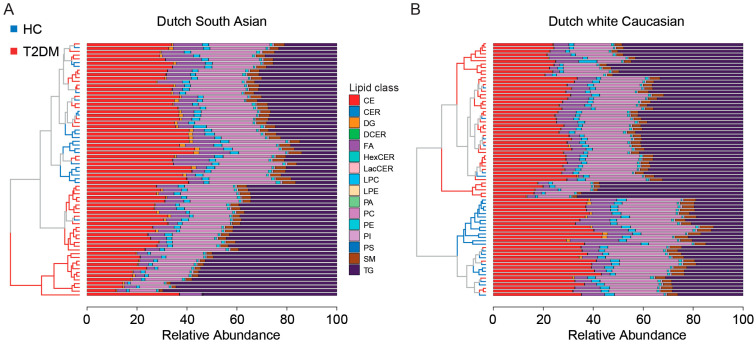
Lipid class abundance between patients with T2DM and healthy controls. (**A**) Stack plot with the hierarchical cluster in Dutch South Asian. (**B**) Stack plot with the hierarchical cluster in Dutch white Caucasian. Abbreviations: CE cholesteryl ester; CER ceramide; DCER dihydroceramide; DG diacylglyceride; FA fatty acid; HC healthy control; HexCER hydroxyceramide; LacCER lactosylceramide; LPC lysophosphatidylcholine; LPE lysophosphatidylethanolamine; PA phosphatidic acid; PC phosphatidylcholine; PE phosphatidylethanolamine; PI phosphatidylinositol; PS phosphatidylserine; SM sphingomyelin; T2DM type 2 diabetes mellitus, TG triglyceride.

**Figure 3 metabolites-14-00033-f003:**
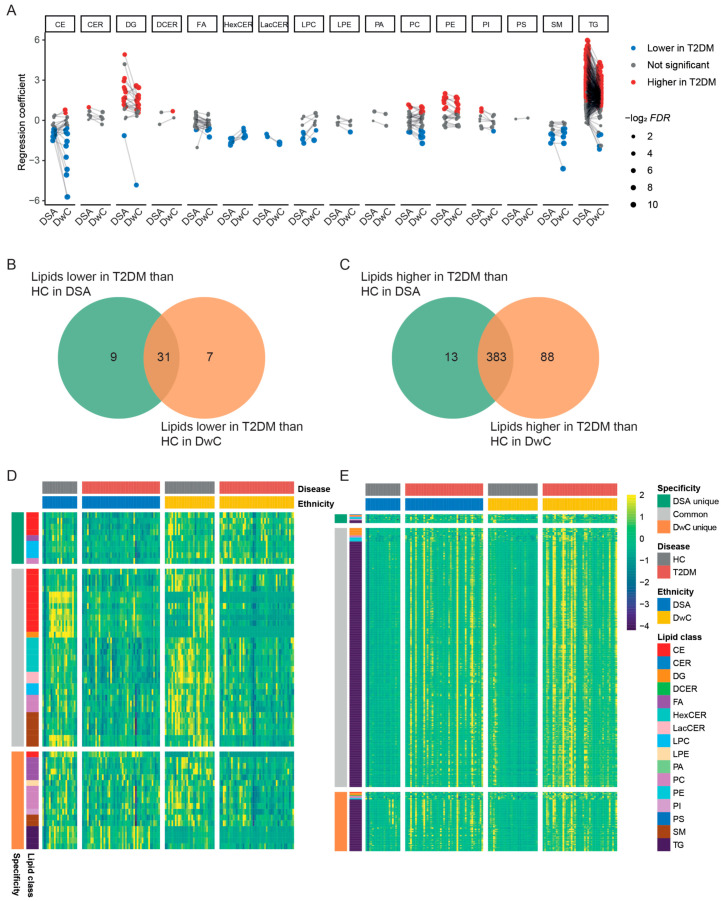
Comparison of differential lipids between patients with T2DM and healthy controls in two ethnicities. (**A**) Differential lipids per lipid class between patients with T2DM and healthy controls, as well as a comparison across two ethnicities. The colour grey represents lipids with non-significant associations (FDR > 0.05), the colour blue represents lipids lower in T2DM, and the colour red represents lipids higher in T2DM. The dot size represents −log_2_FDR. Venn diagram of (**B**) lipids lower in T2DM and (**C**) higher in T2DM than healthy controls (HC) in DSA and DwC. Heatmap of lipids that are commonly/uncommonly (**D**) lower and (**E**) higher in DSA and DwC with T2DM. Abbreviations: CE cholesteryl ester; CER ceramide; DCER dihydroceramide; DG diacylglyceride; DwC Dutch white Caucasian; DSA Dutch South Asian; FA fatty acid; FDR false discovery rate; HexCER hydroxyceramide; LacCER lactosylceramide; LPC lysophosphatidylcholine; LPE lysophosphatidylethanolamine; PA phosphatidic acid; PC phosphatidylcholine; PE phosphatidylethanolamine; PI phosphatidylinositol; PS phosphatidylserine; SM sphingomyelin, TG triglyceride.

**Figure 4 metabolites-14-00033-f004:**
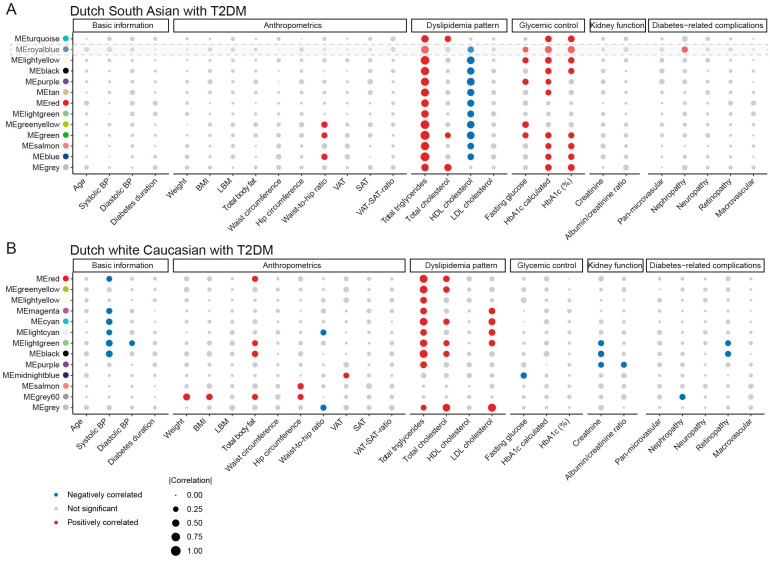
Association of lipid correlation network modules with clinical features in (**A**) Dutch South Asians with T2DM and (**B**) Dutch white Caucasians with T2DM. The colour grey denotes a lipid cluster with no significant associations with clinical features, the colour blue denotes a lipid cluster with a negative association with clinical features, and the colour red denotes a lipid cluster with a positive association with clinical features. The correlation coefficients are represented by the size of the dots (Spearman’s rank correlation test). Abbreviations: BP blood pressure; BMI body mass index; HbA1c hemoglobin A1c; HDL high-density lipoprotein; LBM lean body mass; LDL low-density lipoprotein; SAT subcutaneous adipose tissue, VAT visceral adipose tissue.

**Figure 5 metabolites-14-00033-f005:**
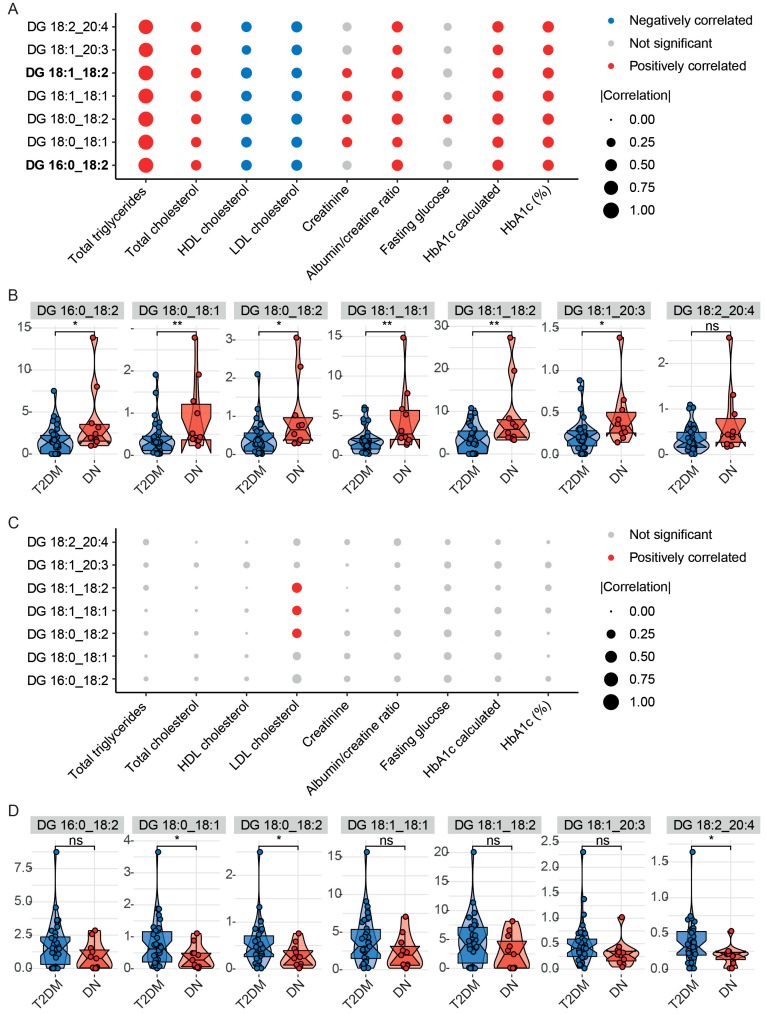
Correlations between key mediatory lipids in diabetic nephropathy-associated module of Dutch South Asians and lipoproteins, kidney function, and glycemic control. (**A**) Bubble plot depicting the correlations of lipids with lipoproteins, kidney function, and glycemic control in Dutch South Asians with T2DM. (**B**) Violin plots of lipids between T2DM with and without diabetic nephropathy in Dutch South Asians. (**C**) Bubble plot depicting the correlations of lipids with lipoproteins, kidney function, and glycemic control in Dutch white Caucasians with T2DM. (**D**) Violin plots of lipids between T2DM with and without DN in Dutch white Caucasians. Lipids in bold indicated that they were specifically different in Dutch South Asians. The colour grey indicates no significant correlations with clinical features; the colour blue indicates a negative correlation with clinical features, and the colour red indicates a positive correlation with clinical features. The size of the dots represents the correlation coefficients (Pearson’s correlation). The Wilcoxon signed-rank test was performed; * *p* < 0.05, ** *p* < 0.01, ns not significant. Abbreviations: DG diacylglyceride; DN diabetic nephropathy; HbA1c hemoglobin A1c; HDL high-density lipoprotein; LDL low-density lipoprotein, T2DM type 2 diabetes mellitus.

## Data Availability

The data presented in this study are available on request from the corresponding author. The data presented in this study, not already enlisted in tables and figures, are not publicly available due to participant privacy.

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
