# Peer review of "Ethnic Disparities in Lipid Metabolism and Clinical Outcomes between Dutch South Asians and Dutch White Caucasians with Type 2 Diabetes Mellitus"

_metabolites, 2024, doi:10.3390/metabo14010033_

Round 1

Reviewer 1 Report

Comments and Suggestions for Authors

This is an interesting study addressing the relationship between lipid profiles, T2DM, and ethnicity.

However, the lipid profile is affected not only by genetics but also by dietary lifestyle. I suggest that the authors add a brief discussion about the influence of differences between diets of Dutch South Asians and Dutch white Caucasians and the lack of information about dietary patterns.

Below, you will find some works that could be consulted, but you can find other authors and works about specific South Asian populations.

https://doi.org/10.3389/fnut.2022.1108098

https://doi.org/10.1093/jn/129.7.1239

https://doi.org/10.3390/ijerph18073328

10.1079/phn2001149

Author Response

This is an interesting study addressing the relationship between lipid profiles, T2DM, and ethnicity.

However, the lipid profile is affected not only by genetics but also by dietary lifestyle. I suggest that the authors add a brief discussion about the influence of differences between diets of Dutch South Asians and Dutch white Caucasians and the lack of information about dietary patterns.

Below, you will find some works that could be consulted, but you can find other authors and works about specific South Asian populations.

https://doi.org/10.3389/fnut.2022.1108098

https://doi.org/10.1093/jn/129.7.1239

https://doi.org/10.3390/ijerph18073328

10.1079/phn2001149

Re: Thank you for your valuable feedback on our study and with the specific references related to the influence of diet on lipid profiles and ethnic populations. We appreciate your suggestion regarding the influence of dietary lifestyle on lipid profiles and its relevance to our study comparing Dutch South Asians and Dutch white Caucasians.

We agree that dietary factors can significantly impact lipid profiles, and considering this aspect would enhance the discussion of our findings. We have included a brief discussion addressing the potential influence of differences between the diets of Dutch South Asians and Dutch white Caucasians on lipid profiles and acknowledged the limitation of lacking specific information about the dietary patterns in our study population. We have incorporated relevant information from these works into our revised manuscript (lines 403-406).

Reviewer 2 Report

Comments and Suggestions for Authors

The Authors presented study about differences in lipidomic profile testing in T2DM patients from different ethnic groups, using NMR method. Although this topic seems to be novel and interesting for the readers, some comments must be addressed.

1. Is the numebr of participants sufficient to obtain reliable results? Can the Authors cofirm this with appropriate statistical tests?

2. Please explain why the following inclusion criteria were used: BMI >= 23, HbA1c 6,5-11%.

3. The presentation of results might be difficult to understand for poteuntial readers. The result could be presented in the form.of tables, with numerical data, which would facilitate interpretation. Please provide p values that demostrate statistical significane.

Author Response

  1. Is the numebr of participants sufficient to obtain reliable results? Can the Authors cofirm this with appropriate statistical tests?

Re: Thank you for your comments. Normally, power analysis is needed, and it is typically performed during the planning phase of a study to determine the required sample size for detecting significant effects or differences between groups. Since our study is a secondary analysis which utilizes existing data from a previous study (Paiman et al. Cardiovasc Diabetol, 2019 and van Eyk et al. Cardiovasc Diabetol, 2019) with study design shown in ClinicalTrials.gov [NCT01761318] and [NCT02660047], respectively (refs 16 and 17), the sample size was already fixed, and it is not feasible or meaningful to perform a power analysis afterwards. Instead, for our analysis, we rely on the available data set and explored the associations or relationships between variables, without specifically assessing statistical power. The goal of our secondary analysis was to derive additional insights or examine specific research questions using the existing dataset. In light of this, we revised our response to highlight the nature of secondary analysis and its reliance on the available data, rather than incorporating information about power analysis.

  1. Please explain why the following inclusion criteria were used: BMI >= 23, HbA1c 6,5-11%.

Re: The inclusion criteria of BMI >= 23 and HbA1c 6.5-11% were selected based on the previous studies and clinical guidelines for diabetes management (Paiman et al. Cardiovasc Diabetol, 2019 and van Eyk et al. Cardiovasc Diabetol, 2019). These ranges represent a commonly used threshold for classifying overweight/obesity and glycemic control in individuals with T2DM.

  1. The presentation of results might be difficult to understand for poteuntial readers. The result could be presented in the form.of tables, with numerical data, which would facilitate interpretation. Please provide p values that demostrate statistical significane.

Re: We understand the concern regarding the presentation of the results. However, if we present the results in form of tables, it will occupy too many pages. Therefore, to improve clarity and ease of interpretation for readers, we provided additional results in the form of tables, including the numerical data as supplementary data (Supplementary table S8-11). Besides, we included exact p-values to demonstrate the statistical significance of our findings.

Reviewer 3 Report

Comments and Suggestions for Authors

1.       Reference to suppl Table S1 is missing in the text

2.       Lipid classes” are mentioned in the text starting from the Introduction, and its definition appears only in line 174

3.       In Line 86, it is written that exclusion criteria was renal disease, but diabetic nephropathy as a complication is widely discussed below. Possibly, some clarification is needed.

4.       External dataset of Chinese IgA nephropathy patients, in my opinion, could not be a validating cohort because of 1) Chinese are not South Asians, they are East Asians, 2) they live in another environment, and, as the authors noted in limitations, 3) it is another disease. According ref [13], Chinese have lipid metabolism different from South Asians (Indians) and whites.

5.       In Line 148, “modules tagged with colour codes” were firstly mentioned, but there is not any description of these (amount, composition of the modules). Is it possible to add this information to supplementary, for example?

There are two colour codes in the paper (for lipid classes according structure and for lipid modules according concentration patterns). Confusion arises when reading the text.

6.       In Line 172, five numbers are correspond to four groups

7.       In Line 192, “…with the greatest relative difference, in DSA with T2DM than in DwC with T2DM”, but according to suppl. Fig 2D there is not any difference in DwC  HC and T2DM. Possibly, more precise wording is needed.

8.       The wordings: 1) Line 223 “…DGs and TGs, contributed more to the risk of T2DM in DSA” and 2)  Line 356 “DG, which contributes to the increased risk in development and progression of T2DM among SAs” seem inaccurate.  According the cited literature, altered DG and TG levels may be not cause the T2DM, but consequences of insulin resistance. For example, in Line 339: “insulin resistance affects regulation of lipid and lipoprotein metabolism [34,35]”, in Line 344 “Insulin resistance also impairs glucose metabolism and TG metabolism [36-38]” . In Line 387, authors write: “we cannot address issues of causality in the association of T2DM”. Possibly, “association” but not “contribution” should be used in this situation.

9.       Word “colour” is used in the paper text, but “color” in Figure legends.

Author Response

Thank you for providing us with your valuable feedback on our manuscript. We greatly appreciate your thorough review, and we have carefully considered each of your points. Below is our response and the actions we will take to address your comments:

  1. Reference to suppl Table S1is missing in the text

Re: We apologize for the oversight regarding the missing reference to Supplementary Table S1 in the text. We will ensure that this reference is appropriately included in the revised manuscript.

  1. Lipid classes” are mentioned in the text starting from the Introduction, and its definition appears only in line 174

Re: We acknowledge that the term "lipid classes" was mentioned earlier in the text without a clear definition. To rectify this, we will clarify the definition of lipid classes in the Materials and Methods section 2.2. Lipidomics profiling using the SLA platform (lines 106-112) to create a more coherent flow throughout the manuscript.

  1. In Line 86, it is written that exclusion criteria was renal disease, but diabetic nephropathy as a complication is widely discussed below. Possibly, some clarification is needed.

Re: We understand the confusion around the term "renal disease" and its relationship to diabetic nephropathy as a complication. We will provide further clarification to emphasize that individuals with pre-existing renal diseases of non-diabetic origin were excluded from the study (line 86), but those with diabetic nephropathy as a specific complication of diabetes were included in the analysis.

  1. External dataset of Chinese IgA nephropathy patients, in my opinion, could not be a validating cohort because of 1) Chinese are not South Asians, they are East Asians, 2) they live in another environment, and, as the authors noted in limitations, 3) it is another disease. According ref [13], Chinese have lipid metabolism different from South Asians (Indians) and whites.

Re: We appreciate your insight regarding the external dataset of Chinese IgA nephropathy patients being not entirely suitable as a validating cohort due to differences in ethnicity and environment compared to South Asians. However, it is important to note that this specific dataset is the only one accessible to us for validating the association between DG lipids and renal function. We have acknowledged this limitation in our manuscript, lines 399-403.

  1. In Line 148, “modules tagged with colour codes” were firstly mentioned, but there is not any description of these (amount, composition of the modules). Is it possible to add this information to supplementary, for example?

There are two colour codes in the paper (for lipid classes according structure and for lipid modules according concentration patterns). Confusion arises when reading the text.

Re: We apologize for the lack of detailed information regarding the modules tagged with colour codes. We rephrased the words in the statistics section to clarify the modules generated from WGCNA analysis (lines 151-152). In the revised version, we also included additional clarification regarding the amount and composition of these modules in the supplementary materials to enhance clarity for readers (see supplementary table S6 and S7).

  1. In Line 172, five numbers are correspond to four groups

Re: Sorry, we don’t understand this comment. In Line 172 of the original manuscript, it is “3.2 Healthy individuals of Dutch South Asian ethnicity reveal a pre-diabetes lipid class profile”.

  1. In Line 192, “…with the greatest relative difference, in DSA with T2DM than in DwC with T2DM”, but according to suppl. Fig 2D there is not any difference in DwC  HC and T2DM. Possibly, more precise wording is needed.

Re: We generated this result from Figure S2E instead of Figure S2D, it appears that the reviewer may have misinterpreted this.

  1. The wordings: 1) Line 223 “…DGs and TGs, contributed more to the risk of T2DM in DSA” and 2)  Line 356 “DG, which contributes to the increased risk in development and progression of T2DM among SAs” seem inaccurate.  According the cited literature, altered DG and TG levels may be not cause the T2DM, but consequences of insulin resistance. For example, in Line 339: “insulin resistance affects regulation of lipid and lipoprotein metabolism [34,35]”, in Line 344 “Insulin resistance also impairs glucose metabolism and TG metabolism [36-38]” . In Line 387, authors write: “we cannot address issues of causality in the association of T2DM”. Possibly, “association” but not “contribution” should be used in this situation.

Re: Upon review, we acknowledge that using the term "contributes" in Lines 223 and 356 of the original manuscript may have been misleading. We agree with your interpretation that altered DG and TG levels are often observed as consequences of insulin resistance rather than direct causes of T2DM. This is supported by the cited literature discussing the impact of insulin resistance on lipid and lipoprotein metabolism, as well as impaired glucose and TG metabolism. To accurately represent the findings and clarify the nature of the association, we revised the wording accordingly. We used terms such as "association" instead of "contribution" to reflect the known associations between altered DG and TG levels and insulin resistance, without implying a direct causal relationship with T2DM (lines 227 and 359).

  1. Word “colour” is used in the paper text, but “color” in Figure legends.

Re: We apologize for the inconsistency in the usage of the terms "colour" and "color." We will ensure that the correct spelling, "colour," is used consistently throughout both the text and figure legends.